# Radon Exposure in the Underground Tourist Route–Historic Silver Mine in Tarnowskie Góry, Poland

**DOI:** 10.3390/ijerph192315778

**Published:** 2022-11-27

**Authors:** Agata Grygier, Krystian Skubacz, Małgorzata Wysocka, Michał Bonczyk, Adam Piech, Mirosław Janik

**Affiliations:** 1Silesian Centre for Environmental Radioactivity, Central Mining Institute (GIG), Plac Gwarków 1, 40166 Katowice, Poland; 2Historic Silver Mine in Tarnowskie Góry, Szczęść Boże 81, 42600 Tarnowskie Góry, Poland; 3National Institute of Radiological Sciences (NIRS), National Institutes for Quantum Science and Technology (QST), Anagawa 4-9-1, Inage, Chiba 263-8555, Japan

**Keywords:** radon exposure, underground show mine, year-round measurements, seasonal variability, track detectors

## Abstract

An assessment of the exposure of workers and tourists to radon in the underground tourist route of the Historic Silver Mine in Tarnowskie Góry was carried out. The study was conducted over a one-year period to capture seasonal variations in radon concentrations. CR-39 track detectors were used to measure radon concentrations, which were exposed in the mine during the following periods: 9 February 2021–19 May 2021, 19 May 2021–26 August 2021, 26 August 2021–25 November 2021 and 25 November 2021–3 March 2022. The annual average radon concentration along the tourist route was 1021 Bq m${}^{-3}$. The highest measured concentration was 2280 Bq m${}^{-3}$ and the lowest concentration was 80 Bq m${}^{-3}$. Based on the measured concentrations, effective doses were calculated, assuming that employees spend 1350 h a year in underground areas and that the time of visiting the mine by tourists is ca. 1 h. The average annual effective dose a worker would receive is approximately 2.5 mSv, and a tourist below 2 μSv. The dose limit expressed as the annual effective dose is 1 mSv for members of the general public and 20 mSv for occupational exposure.

## 1. Introduction

Radon and its short-lived decay products pose a serious threat to human health. Radon, with a half-life of 3.8 days, is a naturally occurring radioactive gas that belongs to the uranium series (${}^{238}$U). A pathogenic factor that increases the likelihood of cancer is the radioactivity of radon, in particular its short-lived decay products ${}^{218}$Po (half-life 3.05 min), ${}^{214}$Pb (half-life 26.8 min), ${}^{214}$Bi (half-life 19.7 min) and ${}^{214}$Po (half-life 164 s) [1,2]. Ludewig and Lorenser in 1924 suggested that radon might have an impact on increased cancer risk [3]. The importance of these risks is reflected in the 2013 European Union Directive, which devotes a large section to radon risk issues at workplaces [4]. According to Polish law, it is necessary to monitor the radon concentration or potential alpha energy concentration of short-lived radon progeny in underground workplaces. The reference level for annual average radon concentration in workplace air is 300 Bq m${}^{-3}$. In turn, the dose limit, expressed as the annual effective dose, is 20 mSv for category A workers and 6 mSv for category B workers [5].

The problem of radon hazard in underground workplaces (especially in operating mines) has been studied for years [6,7]. Mining-related lung cancer was first described by Harting Hesse in 1879 [8]. Since then, much work has been done, and an extensive account of the history of radon, lung cancer, and epidemiological studies can be found elsewhere [9,10,11]. In Poland, there are many underground tourist routes in mines and caves, which is why research was conducted on this subject [12,13,14]. It has been found that in some underground facilities workers receive an annual effective radiation dose exceeding the limit of 20 mSv/year.

Typically, passive solid state nuclear track detectors (SSNTDs) are used for this type of long-term research, or active ones to observe changes in concentrations in the short and medium terms [15,16,17,18,19,20].

The aim of this work was to investigate the radon hazard in the underground tourist route of the Historic Silver Mine in Tarnowskie Góry using passive SSNTD CR-39 type detectors. Seasonal variations in radon concentrations were studied and effective doses for workers as well as tourists were evaluated.

## 2. Material and Methods

### 2.1. Measurement Site

The measurements of radon were performed in a former silver mine, located within the city of Tarnowskie Góry, in the Silesian Upland. The bedrock of Tarnowskie Góry consists of strongly folded Upper Carboniferous, Triassic and Jurassic formations. The Carboniferous, Triassic and Jurassic rocks are overlain by Quaternary sands, silts and clays [21]. The radon risk in the former silver mine is mainly related to the outcrop area of Lower Triassic and Middle Triassic rocks [22]. The oldest written records related to the mining of lead and silver ores, deposited in Middle Triassic dolomites, date to the 12th century. Over the centuries, miners drilled more than 150 kilometres of galleries and 20,000 shafts in the area of Tarnowskie Góry. On 5 September 1976, after many years of operation, the mine was opened for tourists [23]. The current tourist route is a 1740 m long labyrinth connecting the mine’s 3 shafts “Anioł”, “Żmija” and “Szczęść Boże” (Figure 1). The depth of the mine is 40 m. Between the “Szczęść Boże” and “Żmija” shafts, there is a 270 m long water gallery, where tourists can cross the mine by boat.

The temperature in the mine remains constant throughout the year. The average annual temperature is 10.6 °C. The humidity is high (average 90%) and also remains unchanged throughout the year. The average pressure is 996.8 hPa. The ventilation network with marked radon measurement points is shown in Figure 2. The arrows in figure show the direction of the air flow. The ventilation of the mine is carried out in a manner adapted to the schedule of tourists’ visits, i.e., natural (gravitational ventilation) when the mine is closed and forced (mechanical ventilation) when visits take place. Along the entire tourist route open to the public, 30 radon measurement points were set up and two detectors were placed at each measuring point.

### 2.2. Measurement Method

Solid state nuclear track detectors (CR-39) from Radosys (Radosys Ltd., Budapest, Hungary) were used for measurements of radon concentration. The detectors were exposed for a period of three months and then collected for reading and replaced with new ones. Measurements were continued throughout the year, with measurement periods in accordance with the seasons. The first measurement period lasted from 9 February 2021 to 19 May 2021 (Spring), the second one from 19 May 2021 to 26 August 2021 (Summer), the third period from 26 August 2021 to 25 November 2021 (Autumn) and the fourth one from 25 November 2021 to 3 March 2022 (Winter). After exposure, the detectors were transported to the Laboratory of the Silesian Centre for Environmental Radioactivity, where they were chemically treated and analysed. In order to etch the detectors, the CR-39 film on which the tracks of alpha particles were recorded was removed from the diffusion chamber and placed in a cassette, then the detectors were annealed in a 25% NaOH solution at 90 degrees for 3.5 h. In the next step, the number of tracks on the detectors was read using a Radosys Radometer microscope. The applied method enables the lower limit of detection 8 Bq m${}^{-3}$ to be reached for a 3-month measurement period. For the estimation of the uncertainty of the calculated radon concentration, following factors should be taken into consideration: the uncertainty of the track density (10%), the uncertainty of the calibration factor (5%), the uncertainty of the exposure time (6 h), the uncertainty of the track background density (0.1 tracks mm${}^{-2}$). In total, determination of radon activity concentration was made with an uncertainty of about 20% at a confidence level of 0.95.

The concentrations were calculated according to the formula below: (1)$$C_{Rn}=\frac{\left(G_{S}-G_{Sb}\right)}{T}\cdot C_{F}$$
where $C_{Rn}$ is average radon ${}^{222}$Rn concentration [Bq m${}^{-3}$], $G_{S}$ is track density read from the CR-39 foil [tracks mm${}^{-2}$], $G_{Sb}$ is background track density [tracks mm${}^{-2}$], *T* is exposure time [h], CF is calibration factor, determined in the laboratory [(mm${}^{2}$ tracks${}^{-1}$)(Bq h m${}^{-3}$)].

It should be noted that our laboratory is accredited by the Polish Center for Accreditation to perform tests using the method described earlier, which guarantees high quality of measurements. We regularly participate in proficiency tests exercises, obtaining confirmation of the correctness of radon concentration measurements. For example, in 2016 we were one of 14 participants in exercise organized by the Central Mining Institute (GIG) [24]. In 2018 and 2021, we took part in intercomparison of radon measurement organized by the Central Laboratory for Radiation Protection (CLOR) in Warsaw, in which 14 and 11 institutions participated, respectively.

## 3. Results

### 3.1. Radon Concentration

Table 1 shows the radon concentrations for the different measurement periods. The first column contains measurement point id from 1 to 30, the distribution of which is shown in Figure 2. Columns 2 to 5 contain the measured activity concentrations of radon in the mine.

Table 2 shows descriptive statistics of the results obtained for the measurements of radon concentration in all 30 measurement points of the mine for four seasons and the statistic for the annual average radon concentration.

The arithmetic average of radon concentrations (with standard deviation in brackets) in spring, summer, autumn and winter were found as: 858 (411) Bq m${}^{-3}$, 1513 (315) Bq m${}^{-3}$, 1315 (416) Bq m${}^{-3}$ and 397 (337) Bq m${}^{-3}$, respectively. The annual average radon concentration was 1021 (568) Bq m${}^{-3}$. The results showed higher radon concentrations in summer and autumn compared to spring and winter, as presented in Figure 3.

Figure 4 shows the radon concentration at 30 measuring points of the mine in four measuring seasons. The highest radon concentration—2280 Bq m${}^{-3}$—was measured in the summer time (19 May 2021–26 August 2021) in the “Zawałowa” chamber, with the lowest—80 Bq m${}^{-3}$ in the winter time (25 November 2021–3 March 2022) in the “Żmija” downcast shaft, the place where fresh air is supplied from the surface. Similarly, to results reported by other authors, the highest concentrations were measured in summer and the lowest in winter [25,26,27,28]. In winter, the measured concentrations reached the lowest values and ranged from 80–1780 Bq m${}^{-3}$, while in summer they were the highest, from 810 to 2280 Bq m${}^{-3}$. In spring, radon activity concentrations ranged from 160 to 1990 Bq m${}^{-3}$ and in autumn from 530 to 2180 Bq m${}^{-3}$. Radon concentrations, in addition to varying external atmospheric conditions, can also be influenced by many other factors, such as porosity, permeability, uranium content, hydrological conditions [29].

In Figure 4, it can also be seen that the seasonal dynamics or radon change from point to point, but the order the sequence is usually like winter<spring<autumn<summer, or winter<spring<summer<autumn.

Concentration changes at a specific measurement site can be assessed as the ratio of standardized deviation over the entire period to total average radon activity concentration as presented in Figure 5(top). According to the relationship defined as $d=100*\frac{\left(\bar{Rn_{i}}-\bar{Rn}\right)}{\bar{Rn}}$, in %, where i is measurement point, the concentration of radon at points 5, 6, 7, 8, 11, 13, 18, 21, 24, 25, 26 and 30 does not differ, i.e., less than 10%, from the average radon concentration calculated for all 30 measurement points (1021 Bq m${}^{-3}$). However, the relative standard deviations expressed in %, with formula $RSD=100*\frac{\sigma_{Rn_{i}}}{\bar{Rn_{i}}}$, as shown in Figure 5(bottom), are relatively high >30% at these points. It can be concluded that even the average values at these points are close to the annual average, the variation of radon concentration is high. If there is a need to limit costs, these sites could be representative for the assessment of the radiation situation in the facility.

On the other hand, the activity concentration of radon was the most stable inside the “Zawałowa” chamber (point 29, emphasized by light grey) with RSD of 5% (Figure 5, bottom). The measurement site was there located far from the main stream of the ventilation air, which may indicate low air exchange in this area. Not significant changes of radon concentration can be also observed close to the upcast shaft, measurement points 19 and 20, where all air streams are united before discharging the exhaust air to the surface. In contrast to these places, the radon activity concentration varied most strongly near the downcast shaft, measurement points 1 and 2, where fresh air flows from the surface. The variations in this case reached 90%. In addition, interesting results were obtained in the vicinity of “Srebrna” chamber, measurement points 9, 10, 11 and 12. In these points the values of *d* and RSD parameters are relatively low and varied from 10% to 25% and 25–40%, for *d* and RSD, respectively. Thus, the stream of fresh air entering the mine flows along the galleries but does not reach the depths of the chambers. There are no fresh air inlets in the areas of the chambers, they are not ventilated, in addition these areas are large, so radon exhalation and consequently radon concentrations are consistently high. Either the capacity of mechanical or gravity ventilation does not allow the radon concentration to be maintained at 300 Bq m${}^{-3}$.

To check if there is a difference between the seasons, a test of homogeneity of variances was performed in the first step using the Levane test. It has been shown that the variances of radon concentrations between the seasons are homogeneous at *p* = 0.17. Therefore, the inter-season effect of radon concentration was investigated by ANOVA and other statistical tests. The data analysis of all seasons shows that the radon concentration are statistically significantly different between seasons (*t*-test, *p* < 0.05).

Detailed analysis of the relationship between radon concentrations and seasons was conducted by the pairwise comparisons using *t* tests with pooled SD and Tukey multiple comparisons of averages with 95% family-wise confidence level and results are presented in Table 3 and Table 4.

Both tests indicated that average radon concentration between summer and autumn is not statistically different (*p* > 0.05) whereas statistically different between other seasons, i.e., Spring–Autumn, Winter–Autumn, Summer–Spring, Winter–Spring and Winter–Summer (*p* < 0.05).

### 3.2. Dose Estimation

As the results obtained were quite high in comparison to the reference level, it was decided to determine the effective doses that workers and tourists visiting the mine could encounter. For the calculations, it was assumed that a tourist visits the mine once a year and the exposure time is 60 min. In contrast, the working time used in the calculation is 1800 h/year, the nominal annual working time indicated by the Polish regulations for underground mines when the actual working time cannot be determined [30]. The effective doses from radon inhalation were determined according to the following formula [30,31]: (2)$$E=10^{-3}\times k\left(C_{\alpha }+\delta C_{\alpha }-0.1\right)t$$
where *E* is the effective dose (mSv), *k* is the effective dose per exposure expressed as mSv/(mJhm${}^{-3}$), $C_{\alpha }$ is the potential alpha energy concentration (μJm${}^{-3}$), $\delta C_{\alpha }$ is the uncertainty, (μJm${}^{-3}$), and *t* is the working time (h). If the dose was to be estimated on the basis of radon activity concentration measurements, then the equilibrium factor *F* should be additionally considered
(3)$$C_{\alpha }=5.6\times 10^{-3}FC_{Rn}$$
where $C_{Rn}$ is the radon activity concentration (Bq m${}^{-3}$). In this paper, a value of *F* = 0.2, as recommended for mines by the ICRP [32], was adopted.

Table 5 shows the calculated effective doses for workers and tourists. The effective dose determined for a tourist covering the route in 60 min ranges from ca. 1 μSv to 3 μSv, taking into account the uncertainty depending on the season. The annual average dose is equal to ca. 2 μSv. The annual average effective dose for workers is ca. 3 mSv if uncertainty is included (ranging from 1.1 mSv in winter to 4.9 mSv in summer). As the annual value exceeds the legal effective dose limit for the general population (1 mSv in Poland) [5], counter measures should be taken to reduce the dose received by workers. One possible course of action is to reduce the working time in underground areas. Table 6 shows the different working time options so that the doses received do not exceed given limits. In order for a worker to receive an effective dose lower than the dose limit for members of the general public, they should not work more than ca. 540 h per year, to be a category B worker not more than 3240 h, and not more than ca. 10,800 h to not exceed the permitted limit of 20 mSv. Therefore, exceeding the limit of 20 mSv or even 6 mSv is impossible because the total number of hours a year is less than the calculated time limit, and the nominal annual working time is 1800 h.

## 4. Conclusions

The annual average radon concentration measured in the Historic Silver Mine in the period of 9 February 2021–3 March 2022 was 1021 Bq m${}^{-3}$. The reference level for annual average radioactive radon concentration in the air for workplaces and dwellings in Poland is 300 Bq m${}^{-3}$. Seasonal variations in radon concentrations have been noted. The smallest differences in concentrations are found in chambers—areas with reduced air flow. Analysis of the results obtained indicate that air exchange in the former mine is mainly influenced by natural ventilation and that there is a chimney effect in the mine. In the winter months, when the temperature outside is much lower than the temperature inside the mine (differences of up to 20 °C), warm air from inside is pushed outside, leading to a decrease in radon concentration. On the contrary, in the warm months, when the temperature outside is much higher than the temperature inside the mine, a backdraught can be formed in the mine shaft (the lower density of air outside makes the draught of the chimney too low to ventilate the mine). The most important impact of weather conditions on the concentration of radon along the tourist route may indicate a poor share of mechanical ventilation in the route ventilation. However, to confirm this thesis, additional measurements should be carried out with the ventilation on and off. In addition to poor ventilation and variable external atmospheric conditions, high radon concentration can also be influenced by porosity, permeability, uranium content, hydrological and other conditions. As the humidity in the mine is constant, rainfall is not considered to significantly affect radon concentration.

The average dose estimated for a tourist is about 2 μSv/tour. Considering that a tourist visits the mine once a year, this is not much. As far as workers are concerned, the average annual effective dose is 3.3 mSv. This is above the dose limit predicted for members of the general public (1 mSv). However, it should be taken into account that the use of the annual average radon concentration for dose determination may lead to under- or over-estimation of the dose. Furthermore, it should be borne in mind that the doses were calculated for a working time of 1800 h. As can be seen from the time sheets, the employees of the former mine are on underground workstations for much less time (about 500 h/year). Therefore, obtaining such doses seems to be impossible. Ideally, personal monitoring could be introduced for employees to assess the dose due to exposure to radon and its progeny throughout the year.

For comparison: in a tourist coal mine in Upper Silesia, the dose to tourists does not exceed 0.01 μSv/tour. The maximum annual effective dose to workers, calculated on the base of measurements of alpha potential energy concentration of radon decay products, is 0.71 mSv.

Measured radon concentrations exceed the reference level value of 300 Bq m${}^{-3}$. Therefore, some measures should be taken to reduce radon concentration and consequently workers’ exposure to doses from ionising radiation. This can be performed by reducing the working hours of workers, as shown in Table 6, or by increasing the efficiency of the ventilation. Taking into account economic factors, it is not always possible to implement such measures. In that case, it would be necessary to classify the workers in category B (effective dose 1–6 mSv). It is impossible for workers to obtain the effective dose provided for category A of radiation hazard, as they would have to work for 10800 h/year (Table 3). Workers are also not exposed to the effective dose of 6 mSv, the limit for category B, as it corresponds to a working time of 3240 h/year. The exposure assessment on the underground tourist route has been carried out continuously since February 2021 and it is planned to continue the work.

Another way to reduce the radon concentration is increasing the intensity of the ventilation. Mandatory use of protective masks will reduce the risk of receiving increased doses, resulting from exposure to radon and its decay products. P1, P2, or P3 class masks remove 80%, 94% or 99.95% of particles which are smaller than 2 μm (P1) or 0.5 μm (P2, P3). This means that the doses can be reduced approximately 5 times or even more using appropriately selected respiratory track protection means. The systemic reduction of radon activity concentration is in our opinion a better solution. However, it is more complicated and requires higher costs because it involves replacing the fan to increase the ventilation intensity.

To summarise:The measured values of radon concentrations vary in a wide range from 80 to 2280 Bq m${}^{-3}$, with the reference value (300 Bq m${}^{-3}$) being exceeded in most sampling points.Seasonal variability in radon concentration was observed: the lowest values were measured in winter and the highest in summer.The average dose for tourists is 2 µSv/tour and for workers 3.3 mSv, assuming an annual working time of 1800 h.

## Figures and Tables

**Figure 1 ijerph-19-15778-f001:**
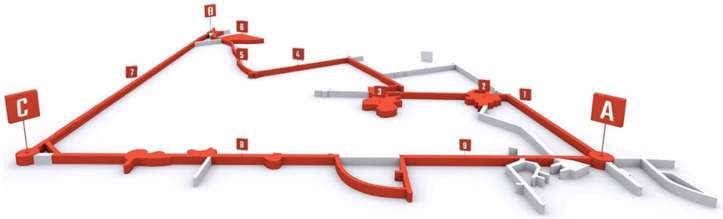
Tourist route in the Historic Silver Mine (A—“Anioł” shaft: B—“Szczęść Boże” shaft; C—“Żmija” shaft; 1—“Staszic” gallery; 2—“Srebrna” chamber; 3—"Zawałowa” chamber; 4—“Niski” gallery; 5—“Wysoki” gallery; 6—“Niska” chamber; 7—water gallery (boat flow); 8—“Okrężny” gallery; 9—“Garus” gallery).

**Figure 2 ijerph-19-15778-f002:**
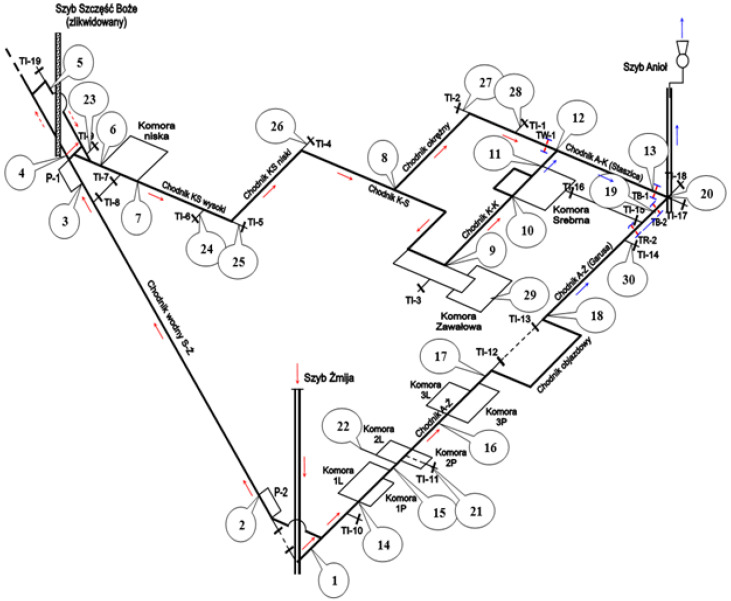
Location of radon measurement points with ventilation network diagram; 1—“Żmija” downcast shaft, 2—inlet to the water gallery; 3—outlet from the water gallery; 4—liquidated and covered shaft “Szczęść Boże”; 5—TI-19 insulating dam; 6—inlet to the “Niska” chamber; 7—outlet from the “Niska” chamber; 8—inlet to the “Okrężny” gallery; 9—inlet to the K-K gallery; 10—inlet to the "Srebrna” chamber; 11—outlet from the “Srebrna” chamber; 12—outlet from the K-K gallery; 13—outlet from the “Staszic” gallery; 14—inlet to the 1L and 1P chambers; 15—inlet to the 2L and 2P chambers; 16—inlet to the 3L and 3P chamber; 17—TI-12 insulating dam; 18—TI-13 insulating dam; 19—outlet from the “Garus” gallery; 20—“Anioł” upcast shaft; 21—TI-11 insulating dam; 22—2L chamber; 23—TI-9 insulating dam; 24—TI-6 insulating dam; 25—TI-5 insulating dam; 26—TI-4 insulating dam; 27—TI-2 insulating dam; 28—TI-1 insulating dam; 29—“Zawałowa” chamber; 30—TI-14 insulating dam. The red arrows mean "fresh air" while the blue ones mean "used air".

**Figure 3 ijerph-19-15778-f003:**
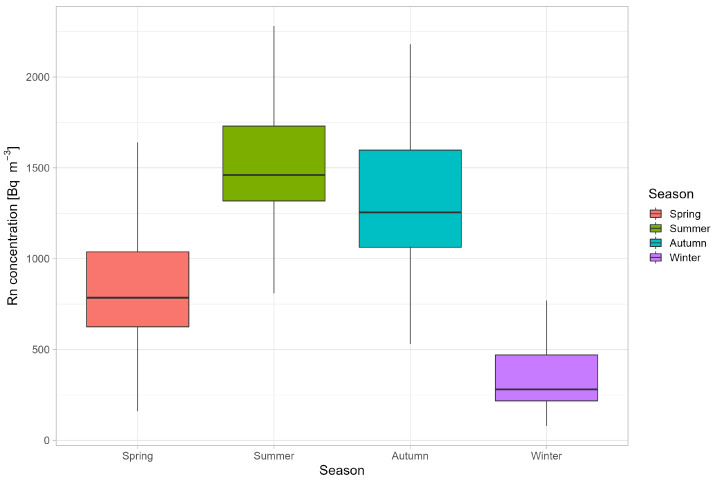
Box plot of seasonal radon concentration for all measurement points.

**Figure 4 ijerph-19-15778-f004:**
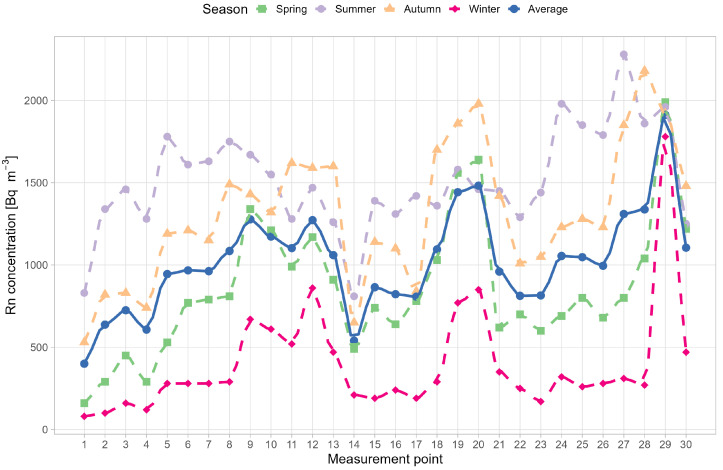
Seasonal variability of radon concentrations in four measurement seasons with average radon concentration (solid line) at 30 measurement points.

**Figure 5 ijerph-19-15778-f005:**
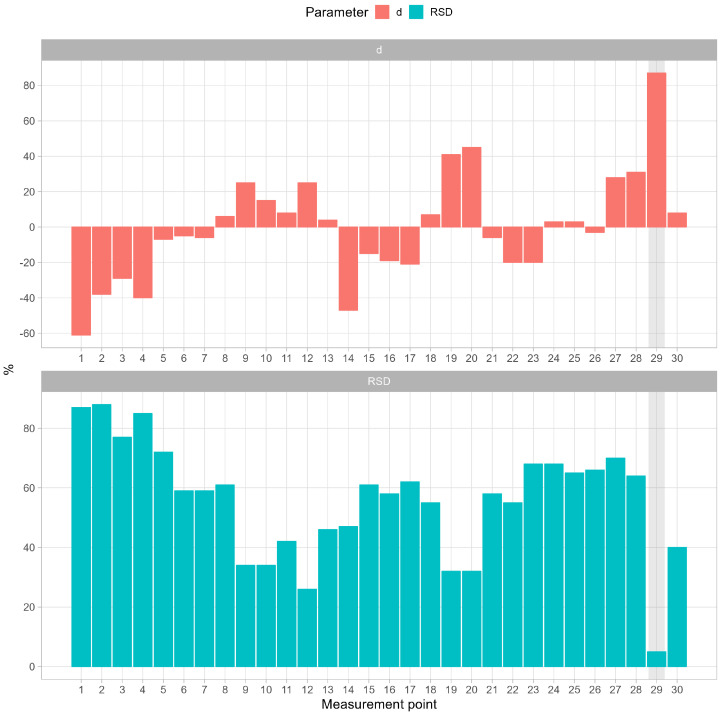
Deviation (d) from the total average radon activity concentration (**top**) and relative standard deviation (RSD) defined as the ratio between standard deviation and average radon concentration for each measurement point (**bottom**).

**Table 1 ijerph-19-15778-t001:** Results of radon activity concentrations ± uncertainty at a 95% confidence level [Bq m${}^{-3}$].

Measurement Point	9 February 2021–19 May 2021 (Spring)	19 May 2021–26 August 2021 (Summer)	26 August 2021–25 November 2021 (Autumn)	25 November 2021–3 March 2022 (Winter)
1.	160 ± 40	830 ± 190	530 ± 130	80 ± 20
2.	290 ± 80	1340 ± 310	820 ± 190	100 ± 20
3.	450 ± 110	1460 ± 330	830 ± 190	160 ± 40
4.	288 ± 29	1280 ± 290	740 ± 170	120 ± 30
5.	530 ± 130	1780 ± 410	1190 ± 270	280 ± 60
6.	770 ± 180	1610 ± 370	1210 ± 280	280 ± 60
7.	790 ± 190	1630 ± 370	1150 ± 260	280 ± 60
8.	810 ± 190	1750 ± 400	1490 ± 340	290 ± 70
9.	1340 ± 310	1670 ± 380	1430 ± 330	670 ± 150
10.	1210 ± 280	1550 ± 350	1320 ± 300	610 ± 140
11.	990 ± 230	1280 ± 290	1620 ± 370	520 ± 120
12.	1170 ± 270	1470 ± 330	1590 ± 360	860 ± 190
13.	910 ± 210	1260 ± 290	1600 ± 360	470 ± 110
14.	490 ± 120	810 ± 190	650 ± 150	210 ± 50
15.	740 ± 170	1390 ± 320	1140 ± 260	190 ± 40
16.	640 ± 150	1310 ± 300	1100 ± 250	240 ± 50
17.	780 ± 180	1420 ± 320	840 ± 190	190 ± 80
18.	1030 ± 240	1360 ± 310	1700 ± 390	290 ± 70
19.	1560 ± 360	1580 ± 360	1860 ± 420	770 ± 170
20.	1640 ± 380	1460 ± 330	1980 ± 450	850 ± 190
21.	620 ± 150	1450 ± 330	1420 ± 330	350 ± 80
22.	700 ± 170	1290 ± 290	1010 ± 230	250 ± 60
23.	600 ± 140	1440 ± 330	1050 ± 240	170 ± 40
24.	690 ± 160	1980 ± 450	1230 ± 280	320 ± 70
25.	800 ± 190	1850 ± 420	1280 ± 290	260 ± 60
26.	680 ± 160	1790 ± 410	1230 ± 280	280 ± 60
27.	800 ± 190	2280 ± 520	1850 ± 420	310 ± 70
28.	1040 ± 240	1860 ± 420	2180 ± 490	270 ± 60
29.	1990 ± 460	1960 ± 450	1920 ± 430	1780 ± 400
30.	1220 ± 280	1250 ± 290	1480 ± 340	470 ± 110

**Table 2 ijerph-19-15778-t002:** Descriptive statistics of radon concentrations according to the measuring season.

Season	Spring	Summer	Autumn	Winter	Annual Average
Median (Bq m${}^{-3}$)	785	1460	1255	280	1035
Arithmetic mean (Bq m${}^{-3}$)	858	1513	1315	397	1021
Standard Deviation (Bq m${}^{-3}$)	411	315	416	337	568
Geometric mean (Bq m${}^{-3}$)	759	1479	1246	313	813
Geometric standard deviation	2	1	1	2	2
Minimum (Bq m${}^{-3}$)	160	810	530	80	80
Maximum (Bq m${}^{-3}$)	1990	2280	2180	1780	2280

**Table 3 ijerph-19-15778-t003:** Results of season comparisons using *t* test with pooled SD (*p* value adjustment method: bonferroni).

Season	Autumn	Spring	Summer
Spring	3.5 × 10${}^{-5}$	-	-
Summer	0.25	2.7 × 10${}^{-9}$	-
Winter	1.7 × 10${}^{-15}$	3.0 × 10${}^{-5}$	<2 × 10${}^{-16}$

**Table 4 ijerph-19-15778-t004:** Results of Tukey multiple comparisons.

Season	Mean Difference	95% Confidence Interval of the Difference (Lower)	95% Confidence Interval of the Difference (Upper)	*p* Value Adjustment
Spring–Autumn	−457.00	−707.64	−206.36	<<0.05
Summer–Autumn	198.33	−52.31	448.97	0.17
Winter–Autumn	−917.33	−1167.97	−666.69	<<0.05
Summer–Spring	655.33	404.69	905.97	<<0.05
Winter–Spring	−460.33	−710.97	−209.69	<<0.05
Winter–Summer	−1115.67	−1366.31	−865.03	<<0.05

**Table 5 ijerph-19-15778-t005:** Effective dose for different exposure period.

Season	Tourists (60 min)	Workers (1800 h)
Spring	1.5 µSv	2.8 mSv
Summer	2.7 µSv	4.9 mSv
Autumn	2.4 µSv	4.3 mSv
Winter	0.6 µSv	1.1 mSv
Average	1.8 µSv	3.3 mSv

**Table 6 ijerph-19-15778-t006:** Calculated working time for dose limits.

Annual Effective Dose [mSv]	Calculated Working Time [h]
1	550
6	3270
20	10,900

## Data Availability

Analyses in this study were based on existing data of radon historic silver mine. The measurements were performed by the authors of the paper and gathered in our database. Data sharing is not applicable to this article.

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
