# Peer review of "Radon Exposure in the Underground Tourist Route–Historic Silver Mine in Tarnowskie Góry, Poland"

_ijerph, 2022, doi:10.3390/ijerph192315778_

Round 1

Reviewer 1 Report

This manuscript deals with the characterization of Radon Exposure in an underground tourist mine in Poland, both to tourists and workers.

It is a really well done and presented work. The Radon risk is properly explained in the introduction along with propper references to previous and related work. The methodology is well explained, providing justification for all the choices selected in the experimental design, enough information on the mine route details is provided and the quality of the activity determinations is assured given that it is performed by an acredited laboratory.

The results section is excellent, all relevant information is provided and structured in a way that makes it easy to check all relevant details. All figures provided are of high quality and allow to check all relevant variations caused by season or location of the measuring points. The variations on the obtained results are properly discussed, relating them to the properties of the ventilation network. The statistical analysis is complete, detailed and properly performed and explained. Finally, the Radon contribution to the dose, both for tourists and workers is provided, proposing mitigation actions for workers.

I only have two small comments/suggestions for the authors:

1. According to the equations provided for the deviation (d) and relative standard deviation (RSD) in lines 115 and 119 respectively, I would expect this quantities to be of order 1. In figure 6 this quantities are instead represented as %. I suggest the authors to modify these equations (by multiplying by 100) such that both the magnitude definitions and their represention represent the same.

2. Line 169 should no be indented.

In summary, I think that this is an excellent and interesting work and I do recommend its publication in  IJERPH, provided that the authors consider my previous suggestions.

Author Response

Dear Reviewer,

Thank you very much for your review, comments, and remarks.

  1. According to the equations provided for the deviation (d) and relative standard deviation (RSD) in lines 115 and 119 respectively, I would expect this quantities to be of order 1. In figure 6 this quantities are instead represented as %. I suggest the authors to modify these equations (by multiplying by 100) such that both the magnitude definitions and their represention represent the same.

Answer: Thank you very much for the advice. This is corrected in the text.

  1. Line 169 should no be indented.

Answer: Corrected.

Reviewer 2 Report

1. Please specify annual average values of SD, minimum and maximum in Table 2.

2. I think that figures 4 and 5 shows the same information. I suggest leaving one of them.

3. A few of references are very old..

Author Response

Dear Reviewer,

Thank you very much for your review, comments, and remarks.

  1. Please specify annual average values of SD, minimum and maximum in Table 2.

Answer: Table 2, in the last column, shows the statistics (SD, minimum, maximum, AM, GM and median) for the results obtained for the whole year.

  1. I think that figures 4 and 5shows the same  I suggest leaving one of them.

Answer:  We decided to remove Figure 5, in Figure 4 we added a series of data with average values so as not to lose the data that were presented in Figure 5.

  1. A few of references are very old.

Answer: We understand that some of the references are old, but we believe they are crucial to the discussion of radon measurements. However, we have added the latest references to the same topic, i.e. radon measurements in underground sites.

Reviewer 3 Report

In this paper, a study of radon exposure in a silver mine is carried out. For this, measurements are made for a year at a sufficient number of sampling points using CR-39 passive detectors.

The paper is correctly written, the scientific methodology is well founded and correctly carried out. The results obtained are correctly discussed and may be of great interest for the evaluation of the radiological protection of workers and the general public in this type of enclosure.

For all these reasons, I consider that the paper can be published in its current form.

Author Response

Dear Reviewer,

Thank you very much for your review

Reviewer 4 Report

The authors provide an interesting work in terms of a radilogical study in an area with high touristic activity. 

The introduction is poor. The authors should include more paper works for using CR39 detectors for radon measurement. Some examples related to the use of such detectors and method analysis are: 

Nuclear Instruments and Methods in Physics Research, Section B: Beam Interactions with Materials and Atoms 267(14), pp. 2440-2448

Computer Physics Communications 177(3), pp. 329-338

Methodology has to be improved according to the international standards. What about calibration and inter-comparison exercises?

Uncertainty budget must be described. 

Figure 4 or 5 is not required since it provided the same info.

The conclusions must be highlighted. 

Author Response

Dear Reviewer,

Thank you very much for your effort

  1. The introduction is poor. The authors should include more paper works for using CR39 detectors for radon measurement.

Answer: References have been added and the text has been modified

  1. Methodology has to be improved according to the international standards. What about calibration and inter-comparison exercises?

Answer: As mentioned in the text (line 77), the laboratory is accredited by the Polish Centre for Accreditation to carry out tests using the method described, thus ensuring high quality testing. Information about participation in comparative measurements was added in the text.

  1. Uncertainty budget must be described. 

Answer: The uncertainty budget is described in the text under methods (line 83 and following).

  1. Figure 4 or 5 is not required since it provided the same info.

Answer: We decided to remove Figure 5, in Figure 4 we added a series of data with average values so as not to lose the data that were presented in Figure 5.

  1. The conclusions must be highlighted. 

Answer: The conclusions are highlighted in the last paragraph.

Round 2

Reviewer 4 Report

the authors improved the manuscript. The should make a proofreading before publication.